# Evaluation of the therapeutic efficiency and efficacy of blood purification in the treatment of severe acute pancreatitis

Hongwei Huang[1], Zhongshi Huang[2], Menghua Chen[3], Ken Okamoto[4]*

1 Department of Intensive Care Medicine, Guangxi Hospital Division of The First Affiliated Hospital, Sun Yat-sen University, Nanning, Guangxi, China, 2 Department of Intensive Care Medicine, Youjiang Medical College for Nationalities Affiliated Hospital, Baise, Guangxi, China, 3 Department of Intensive Care Medicine, The Second Affiliated Hospital of Guangxi Medical University, Nanning, Guangxi, China, 4 Emergency and Intensive Care Unit, Juntendo University Urayasu Hospital, Urayasu, Japan

* Okamoto_qq@163.com.

**Data Availability Statement:** All relevant data are within the manuscript.

**Funding:** The author(s) received no specific funding for this work.

## Abstract

This study aimed to evaluate the therapeutic efficacy and effect of blood purification (BP) therapy on severe acute pancreatitis (SAP). Information on 305 patients (BP group 68, control group 237) diagnosed with SAP was retrieved from the Medical Information Mart for Intensive Care IV (MIMIC IV) database. Firstly, the influence of BP treatment was preliminarily evaluated by comparing the outcome indicators of the two groups. Secondly, multiple regression analysis was used to screen the mortality risk factors to verify the impact of BP on the survival outcome of patients. Then, the effect of BP treatment was re-validated with baseline data. Finally, cox regression was used to make the survival curve after matching to confirm whether BP could affect the death outcome. The results indicated that the BP group had a lower incidence of shock ($p = 0.012$), but a higher incidence of acute kidney injury (AKI) ($p < 0.001$), with no differences observed in other outcome indicators when compared to the control group. It was also found that the 28-day survival curve of patients between the two groups was significantly overlapped ($p = 0.133$), indicating that BP treatment had no significant effect on the survival outcome of patients with SAP. Although BP is beneficial in stabilizing hemodynamics, it has no effect on short- and long-term mortality of patients. The application of this technology in the treatment of SAP should be done with caution until appropriate BP treatment methods are developed, particularly for patients who are not able to adapt to renal replacement therapy.

## Introduction

Severe acute pancreatitis (SAP) is a common acute abdominal disease with many complications and high mortality. It promotes the onset of systemic inflammatory response syndrome (SIRS), necrosis, infection of pancreatic and peripancreatic tissues, and leads to multiple organ dysfunction syndrome (MODS) [1]. Acute kidney injury (AKI) is one of the most common complications of SAP, leading to a dramatic increase in SAP-related mortality [2]. At present,

**Competing interests:** The authors have declared that no competing interests exist.

the incidence of SAP is on the rise worldwide, which puts a great burden on healthcare services. Furthermore, despite the significant advancements made in SAP management, the illness's discomfort, high death rate, lengthy hospital stays, high medical expenses, and low rate of recovery during treatment still pose significant challenges for the medical community [3, 4]. The main form of treatment for SAP is supportive nursing treatment, as no effective drugs have been developed to date.

Blood purification (BP) has steadily grown in importance as a treatment for SAP because it can be used to regularly or sporadically eliminate the body of excess water and solutes, preserve hemodynamic stability, eliminate inflammatory mediators, control immune response, and lessen damage to organ function [5–7]. As one of the treatments for SAP, BP was first used in clinical practice in the 1970s [8]. Some studies have shown that BP treatment can reduce the clinical symptoms of SAP in patients, shorten the length of hospital stay, and improve the patient prognosis [9–11]. However, its efficacy, particularly in terms of long-term survival benefits, is still controversial, and the treatment is still in the exploratory phase [12, 13]. As a result, some international guidelines for SAP management only recommend it very conservatively [14, 15], on the other hand, BP therapy is widely adopted in China. As reported in our previously published meta-analysis, BP therapy, with the exception of high-volume hemofiltration (HVHF) treatment modalities, did not improve the 6-month mortality rate even with early intervention and resulted in longer hospital stays and higher hospitalization costs compared to controls in China [16].

The Medical Information Mart for Intensive Care IV (MIMIC IV) database provided by Beth Israel Deaconess Medical Center (Boston, Massachusetts, USA) is a large publicly accessible database. It stores pertinent information about patients who were admitted to Beth Israel Deaconess Medical Center's intensive care unit (ICU) between 2001 and 2019. The information includes medical and health data from more than 50,000 patients, including records of their survival within a year of discharge. In particular, the database contains demographic information on patients, laboratory test results, medication information, imaging reports, hospitalization records, and nursing levels [17]. Therefore, we sought to determine the therapeutic efficacy and effect of BP therapy in the treatment of SAP abroad by analyzing the MIMIC IV database.

## Materials and methods

### Data resources and selection criteria

The author's retrieval ID (41737357) was obtained after an official assessment by MIMIC. The database was downloaded via PhysioNet according to the instructions (https://mimic.physionet.org). Data on patients diagnosed with acute pancreatitis (AP) prior to January 30, 2022, were retrieved from the MIMIC-IV database (ICD-9 code: 5770, K85) and then merged, screened, and selected using STATA MP 17.0 software.

The inclusion criteria were as follows: (1) Patients whose diagnosis in the database meets the 2012 Atlanta SAP classification criteria [18]; (2) The maximum acute physiology and chronic health evaluation (APACHE) II score during hospitalization is > 8 points, and the sequential organ failure assessment (SOFA) score is > 2 points, with a duration of more than 48 h; (3) Age over 18 years at the time of illness. The exclusion criteria were as follows: (1) Records with a time interval of less than one year since the last hospitalization; (2) Patients with malignant tumors, severe liver disease, baseline creatinine level $\geq$ 450 μmol/L, chronic pancreatitis, or HIV/AIDS. Ethics statement is not applicable to this study.

## Grouping

Two protocols were used to group the selected patients. The first is to classify all patients according to whether they received BP treatment or not into two groups: the BP treatment group and the control group. The other is to group all patients into the absolute indication group, the relative indication group, and the non-indication group according to the indications for BP intervention (KDOQI 2012) [19]. The specific characteristics of the absolute indication group included plasma urea nitrogen concentration levels > 36 mmol/L, uremic encephalopathy, uremic pericarditis, uremic neuropathy and muscle damage; blood potassium > 6.5 mmol/L, blood magnesium > 4 mmol/L; acidosis with pH < 7.15; oliguria < 200 ml/24h or anuria; cerebral and pulmonary edema due to fluid overload; and blood creatinine > 353.5 μmol/L (which increased more than 3 times from the baseline). The relative indication group wa defined by blood creatinine levels of 176.8–353.5 μmol/L (which were increased more than 2 times from the baseline), urine output < 0.5 ml/kg.h for more than 12 h. While the non-indication group was defined as having no absolute or relative indication. Different time points were analyzed to determine whether BP treatment was beneficial or not. The grouping strategy is shown in Fig 1.

## Data collection and extraction

Two authors independently extracted data from the MIMIC IV database. The following data were collected and evaluated in this study: (1) Demographic information: age, sex, height, weight, race, etiology of AP, hospital admission and discharge time, mortality within one year and time of death, total hospital stay, ICU stay, Charlson comorbidity index (CCI), comorbidities (such as liver disease, heart disease, stroke, diabetes, etc.). (2) Laboratory test data: blood test, electrolytes, liver and kidney function, arterial blood gas analysis, and various coagulation indicators. The maximum and minimum values, and values on the first day of ICU admission were retrieved. (3) Oxygen therapy method (including the use of mechanical ventilation), use of vasoactive drugs, and occurrence of AKI. (4) Organ function scores: SOFA score on the first day of ICU admission, and APACHE II score (not included in the database, were calculated based on the relevant data and scoring criteria). (5) Parameters related to BP therapy: whether BP was performed as well as the parameters of BP, including the methods of BP, duration of treatment, anticoagulation, dosage, etc.

## Outcome measures

To evaluate the effect and efficacy of BP in treating SAP, we primarily focused on the following parameters: 28-day and 1-year all-cause mortality, the total length of hospital stays and ICU stays. Mortality rate reflects efficacy, while hospital stays and costs represent efficiency. In addition, the incidence of shock, acute respiratory distress syndrome (ARDS), and AKI were used as secondary outcome measures.

## Statistical analysis

Measurement data in normal distribution were expressed as mean ± standard deviation (SD) and analyzed with Student's $t$-test or analysis of variance. Otherwise, variables were described as medians and interquartile ranges and analyzed by the Mann-Whitney U test or Kruskal-Wallis test.

For Group 1, we first compared the outcome measures of the control group and the BP treatment group to evaluate the preliminary effect of BP on the outcome measures. Secondly, we adopted multivariate logistic regression analysis to identify risk factors for death and

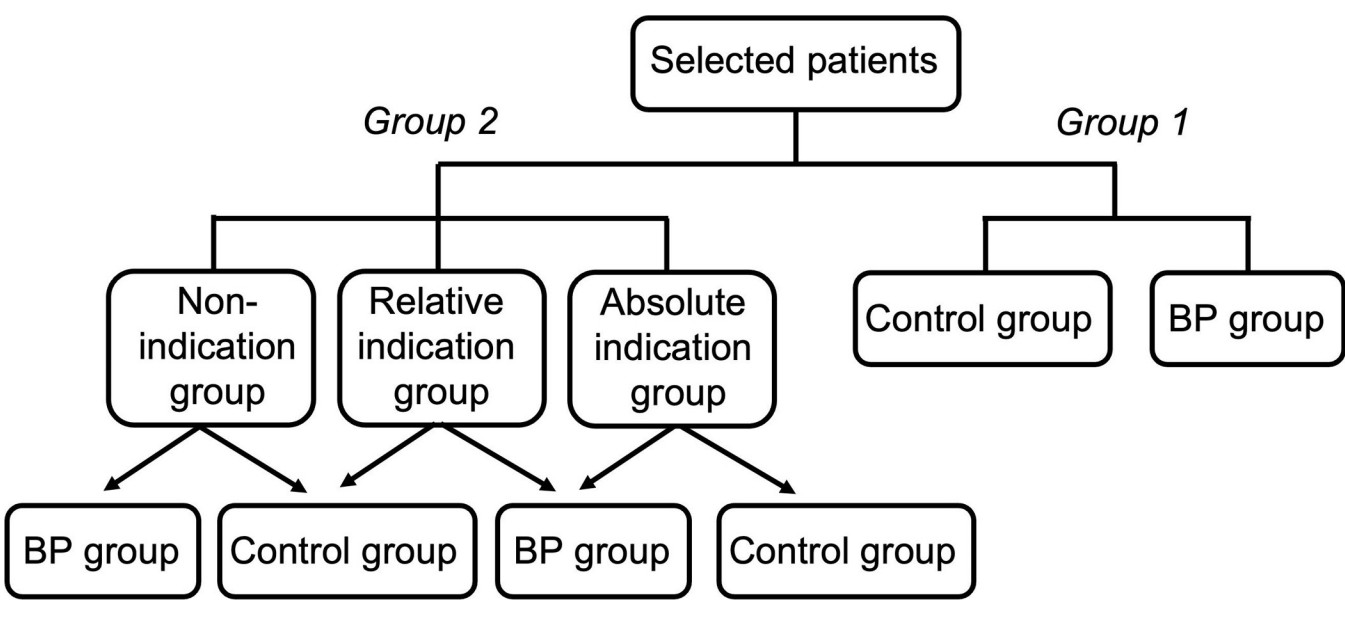

**Fig 1. Groups and subgroups.**

establish a regression equation for the factors affecting death using model correction to verify whether BP can impact patient survival outcomes. Then, propensity score matching with demographic data and death risk factors on the first day of admission was used as baseline data, followed by matched analysis between the two groups to compare the outcome measures and validate the effect of BP on the outcome measures. Finally, Cox regression was used to

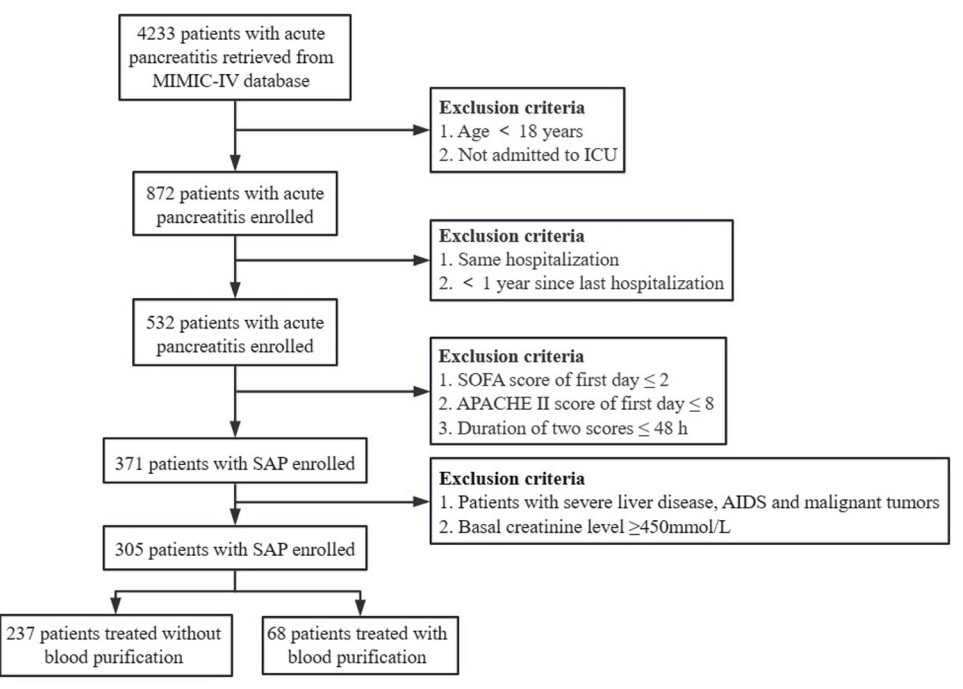

**Fig 2. Process of case screening.**

generate survival curves for the matched data and confirm the effect of BP on mortality outcome. For Group 2, demographic data and death risk factors on the first day of admission were used as baseline data. Propensity score matching was performed, followed by matched analysis to confirm the effect of BP on outcome measures. Chi-square test, *t*-test, and rank sum test were used in this study. SPSS 25.0 and STATA/MP 17.0 software were used for statistical analysis, with a two-sided test and $p < 0.05$ indicating statistical significance.

## Results

### Baseline characteristics and outcome measures of enrolled patients

A total of 4,233 cases were retrieved. According to the inclusion and exclusion criteria, 305 cases (145 males and 160 females) were included after screening (Fig 2). The average age and body mass index (BMI) were 61.1 ± 14.6 years and 28.2 ± 6.5 respectively. Whites made up 53.4% of all included cases, followed by Blacks (30.5%), Hispanics (6.9%), Asians (2.0%), and others (7.2%). The Etiology of AP included cholelithiasis (72 cases), alcohol abuse (158 cases), hypertriglyceridemia (64 cases), and others (11 cases). Most patients had comorbid diseases, with a Charlson comorbidity index score of 5 (IQR 3–7). Specifically, 31.4% had chronic kidney disease (CKD stages 1–4), 28.4% had diabetes, 24.3% had chronic lung disease, and 47.5% had cardiovascular disease (CVD) [chronic heart failure (New York Heart Association class 1–30, hypertension, coronary heart disease]. On the first day of ICU admission, the APACHE II score was 17 (IQR 14–20) and the SOFA score was 6 (IQR 4–9). Of the 305 patients, 68 received BP treatment, while 237 did not. Of the 68 patients that received BP treatment, 44 underwent intermittent hemodialysis (IHD), 9 underwent continuous veno-venous hemofiltration (CVVH), while the rest of the 15 patients underwent unknown BP methods of treatment. The data on anticoagulation, replacement fluid volume, start time of BP, and other information were missing. In addition, there was no information regarding hospitalization costs. The relevant information is shown in Table 1.

The demographic characteristics, first-day SOFA score, APACHE II score, and primary and secondary outcome measures of the two groups are shown in Table 1. Apart from the higher incidence of AKI observed in the blood purification group than in the control group, there was no difference in all other outcome indicators between the groups. However, the baseline data of the two groups were not consistent. The comorbidity index, SOFA score, and APACHE II score on the first day of ICU admission in the control group were lower than those of the blood purification treatment group.

### Risk factors associated with patient mortality

Multiple regression analysis was used to screen for risk factors for mortality (backward method, $p = 0.1$). Model calibration was performed by adding additional factors to the previous model. Specifically, model 1 included patient baseline data and BP treatment. SOFA and APACHE II scores on the first day of admission were added in model 2. Blood cell classification examination was added in model 3. Blood biochemical examination (liver and kidney function, electrolytes, blood gas analysis, and lactate) was added in model 4. Outcome indicators during hospitalization (AKI, ARDS, shock, and GCS score) were added in model 5. The variance inflation factor (VIF), variables, and the probability of BP treatment being eliminated from each model are shown in Table 2. The goodness-of-fit of model 5 was 91.3%. Consequently, age, sex, combined digestive ulcer, maximum lactate value, minimum lymphocyte value, severity of ARDS, and minimum neutrophil value were selected to construct the regression equation, $Y = 0.95 + 1.08X1 + 2.51X2 + 6.98X3 + 1.26X4 + 0.98X5 + 1.745X6 + 1.01X7$ ($Y$ = death risk, $X1$ = age, $X2$ = sex, $X3$ = combined digestive ulcer, $X4$ = maximum lactate

**Table 1. The baseline and outcomes of the two groups.**

| Variables | Control group (n = 237) | BP group (n = 68) | p |
|---|---|---|---|
| **Baseline** | | | |
| Age (years) | 61.1 ± 14.9 | 60.7 ± 13.5 | 0.838 |
| Gender (Male) | 114 (48.1%) | 31 (45.6%) | 0.715 |
| CCI | 4 (IQR 3–7) | 6 (IQR 5–8) | < 0.001 |
| SOFA | 6 (IQR 4–8) | 8 (IQR 6–11) | < 0.001 |
| Etiology of AP | | | 0.230 |
| Cholelithiasis | 56 | 16 | |
| Alcohol abuse | 118 | 40 | |
| Hypertriglyceridemia | 55 | 9 | |
| Others | 8 | 3 | |
| APACHE II | 16 (IQR 13–19) | 19 (IQR 16–23) | < 0.001 |
| **Outcomes** | | | |
| 28-day all-cause mortality | 17 | 9 | 0.115 |
| 1-year all-cause mortality | 25 | 12 | 0.114 |
| Hospitalization time | 8.7 (IQR 4.8–18.0) | 10.2 (IQR 5.0–20.2) | 0.473 |
| ICU time | 2.3 (IQR 1.3–4.8) | 3.4 (IQR 1.3–9.0) | 0.135 |
| AKI | 38,16,4,24[a] | 9,13,9,30[a] | < 0.001 |
| ARDS | 50 | 13 | 0.722 |
| Shock | 92 | 26 | 0.931 |

value, X5 = minimum lymphocyte value, X6 = severity of ARDS, X7 = minimum neutrophil value). Except for model 1, BP treatment was eliminated from the models, suggesting that BP treatment had no impact on mortality.

## Effect of BP on outcome measures

Table 3 indicates that there were no statistically significant differences in death-related factors between the two groups, with the exception of the SOFA and APACHE II scores on the first day of ICU admission. These results suggest that BP treatment had no significant impact on the primary outcome measures of SAP patients. Notably, we observed that the severity of the initial illness was different between the two groups, as evidenced by significant differences in organ function scores on the first day of hospitalization. Therefore, propensity score matching (PSM) was performed based on the patient's baseline data (gender, age, Charlson's comorbidity index, presence of peptic ulcer disease, and SOFA and APACHE II scores on the first day of

**Table 2. Pooled models of multiple factors logistic regression about mortality.**

| Model | VIF | Risk factors | β | RRT exclusion probability |
|---|---|---|---|---|
| Model 1 | 1.26 | Age、RRT | 1.03, 1.93 | 0 |
| Model 2 | 3.93 | Age、SOFA | 1.03, 1.21 | 0.426 |
| Model 3 | 2.77 | Age、Gender、Minimum lymphocyte value、Ulcer、SOFA | 1.07, 2.40, 0.98, 6.44, 1.20 | 0.352 |
| Model 4 | 3.1 | Age、Maximum lactic acid、Ulcer、SOFA、Minimum lymphocyte value、Minimum neutrophil value | 1.05, 1.28, 7.89, 1.16, 0.99, 1.0 | 0.3 |
| Model 5 | 2.35 | Age、Gender、Maximum lactic acid、Ulcer、Minimum lymphocyte value、ARDS、Minimum neutrophil value | 1.08, 2.51, 1.26, 7.19, 0.98, 1.74, 1.0 | 0.16 |

**Table 3. Comparison of mortality factors between two groups.**

| Variable | Control group (n = 237) | BP group (n = 68) | p |
|---|---|---|---|
| Total death | 25 | 12 | 0.114 |
| 28-day all-cause mortality | 17 | 9 | 0.115 |
| 1-year all-cause mortality | 25 | 12 | 0.114 |
| Hospitalization time | 8.7 (IQR 4.8–18.0) | 10.2 (IQR 5.0–20.2) | 0.473 |
| ICU time | 2.3 (IQR 1.3–4.8) | 3.4 (IQR 1.3–9.0) | 0.135 |
| AKI | 38,16,4,24[a] | 9,13,9,30[a] | < 0.001 |
| Minimum neutrophil value ($10^9$/L) | 5.87 (IQR 1.92–11.50) | 6.28 (IQR 3.95–11.46) | 0.599 |
| Minimum lymphocyte value ($10^9$/L) | 0.74 (IQR 0.21–1.29) | 0.75 (IQR 0.35–1.29) | 0.523 |
| Maximum lactic acid | 2.86 (IQR 1.70–3.29) | 2.91 (IQR 1.77–3.33) | 0.606 |
| ARDS | 50 | 13 | 0.722 |
| Shock | 92 | 26 | 0.931 |

a, AKI 1, 2, 3 and acute exacerbation of chronic renal insufficiency.

admission). A total of 62 pairs of cases were successfully matched. Table 4 shows the distribution of baseline variables after matching, while Table 5 shows the outcome indicators of the two groups after matching. There were 118 cases of septic shock reported; the results indicated that the BP group experienced a lower incidence of shock but a higher incidence of acute exacerbation of chronic kidney dysfunction and AKI stage 2–3. There were no differences in other outcome measures between the two groups.

## Effect of BP on survival outcomes

Next, we evaluated the impact of BP treatment on the survival outcomes of 124 matched SAP patients (62 matched pairs). As shown in Fig 3, the 28-day survival curves of the two groups overlapped significantly ($p = 0.133$), indicating that BP treatment had no significant impacts on the survival outcomes of SAP patients.

## Effect of BP methods on outcome measures

Among all 68 patients receiving BP treatment, 44 underwent IHD, 9 underwent CVVH, and the method of BP was unknown for 15 patients. As a result, examining how various BP techniques affected outcome measures was not achievable.

**Table 4. Baseline characteristics of the two groups before and after matching.**

| Variable | Before matching | | | After matching | | |
|---|---|---|---|---|---|---|
| | Control group (n = 237) | BP group (n = 68) | p | Control group (n = 62) | BP group (n = 62) | p |
| Age (years) | 61.1 ± 14.9 | 60.7 ± 13.5 | 0.838[a] | 60.8 ± 12.7 | 61.1 ± 13.9 | 0.925 |
| Gender (Male) | 114 | 31 | 0.715[b] | 30 | 28 | 0.772 |
| CCI | 4 (IQR 3–7) | 6 (IQR 5–8) | < 0.001[d] | 7 (IQR 4–8) | 6 (IQR 5–8) | 0.848 |
| SOFA | 6 (IQR 4–8) | 8 (IQR 6–11) | < 0.001[d] | 8 (IQR 6–11) | 8 (IQR 6–10) | 0.527 |
| APACH II | 16 (IQR 13–19) | 19 (IQR 16–23) | < 0.001[d] | 19 (IQR 16–25) | 18 (IQR 16–21) | 0.365 |
| Ulcer | 11 | 3 | 1[c] | 2 | 2 | 1 |

a, *t*-test of two independent samples; b, Chi-square test, c, Fisher's exact test; d, Rank sum test.

**Table 5. Outcome indicators of the two groups after PSM.**

| Variable | Control group (n = 62) | BP group (n = 62) | p |
|---|---|---|---|
| Total death | 9 | 11 | 0.114[a] |
| 28-day all-cause mortality | 7 | 8 | 0.783[b] |
| 1-year all-cause mortality | 8 | 11 | 0.455[b] |
| Hospitalization time | 9.6 (IQR 5.1–17.3) | 9.6 (IQR 4.9–18.1) | 0.974[c] |
| ICU time | 3.0 (IQR 1.6–4.9) | 3.2 (IQR 1.3–5.6) | 0.865[c] |
| AKI | 11, 5, 3, 14[a] | 9, 13, 6, 27[a] | < 0.001[b] |
| ARDS | 18 | 11 | 0.138[b] |
| Shock | 35 | 21 | 0.012[b] |
| Minimum neutrophil value ($10^9$/L) | 0.45 (IQR 0.16–1.01) | 0.8 (IQR 0.39–1.35) | 0.015[c] |
| Minimum lymphocyte value ($10^9$/L) | 5.68 (IQR 1.24–11.8) | 6.50 (IQR 4.70–12.47) | 0.26[c] |

a, AKI 1, 2, 3 and acute exacerbation of chronic renal insufficiency; b, Chi-square test; c, Rank sum test.

## Effect of BP on patients with renal replacement therapy

Due to the missing data on patients' urine output and imaging, and the lack of records, it was difficult to group patients according to indications for renal replacement therapy (RRT) based on the literature. Therefore, appropriate analyses could not be conducted.

## Discussion

We retrieved data from the MIMIC-IV database followed by performing a propensity score matching analysis,we found that the BP group failed to show benefits in terms of 28-day and 1-year survival compared to the control group. This result is congruent with the conclusion of the second part of the study which reveals the poor efficacy of BP treatment in SAP. There was no difference in hospitalization time or length of ICU stay between the two groups, further validating its poor effeciency.

It was challenging to determine whether there were absolute indications for BP treatment because the extracted data lacked many pertinent parameters (such as treatment dosage, anticoagulation methods, etc.), and it was impossible to access the patient's medical records and imaging data Consequently, analyses were not conducted based on indications and models as done in previous studies. However, conclusions from both sets of data indicate that BP did not improve the survival outcomes of SAP patients. At the same time, it should be noted that there was a large difference in baseline data between the two groups, and the incidence of comorbid diseases in selected patients was high. About 64.7% of the BP group had chronic kidney disease (compared to 22.4% in the control group), and 30 patients had an acute worsening of chronic kidney function (defined as a serum creatinine increase of more than 1.5 times above the baseline) underwent BP treatment (24 people did not receive BP). Presumably, AKI is the main complication of SAP [20], and renal dysfunction or deterioration may lead to the requirement of BP treatment application in these patients. Among the 175 patients who had no obvious indications, only 11 (7.33%) underwent BP, which is in sharp contrast to the 49.4% (41/83) in the second part, these findings indicate that BP treatment for SAP was performed with caution in the United States. The reasons for the difference in enthusiasm for BP treatment between China and the United States are difficult to discern. However, the majority of studies on the use of BP treatment to lower or regulate the concentration of inflammatory mediators and improve clinical symptoms in SAP patients were carried out in China, according to the

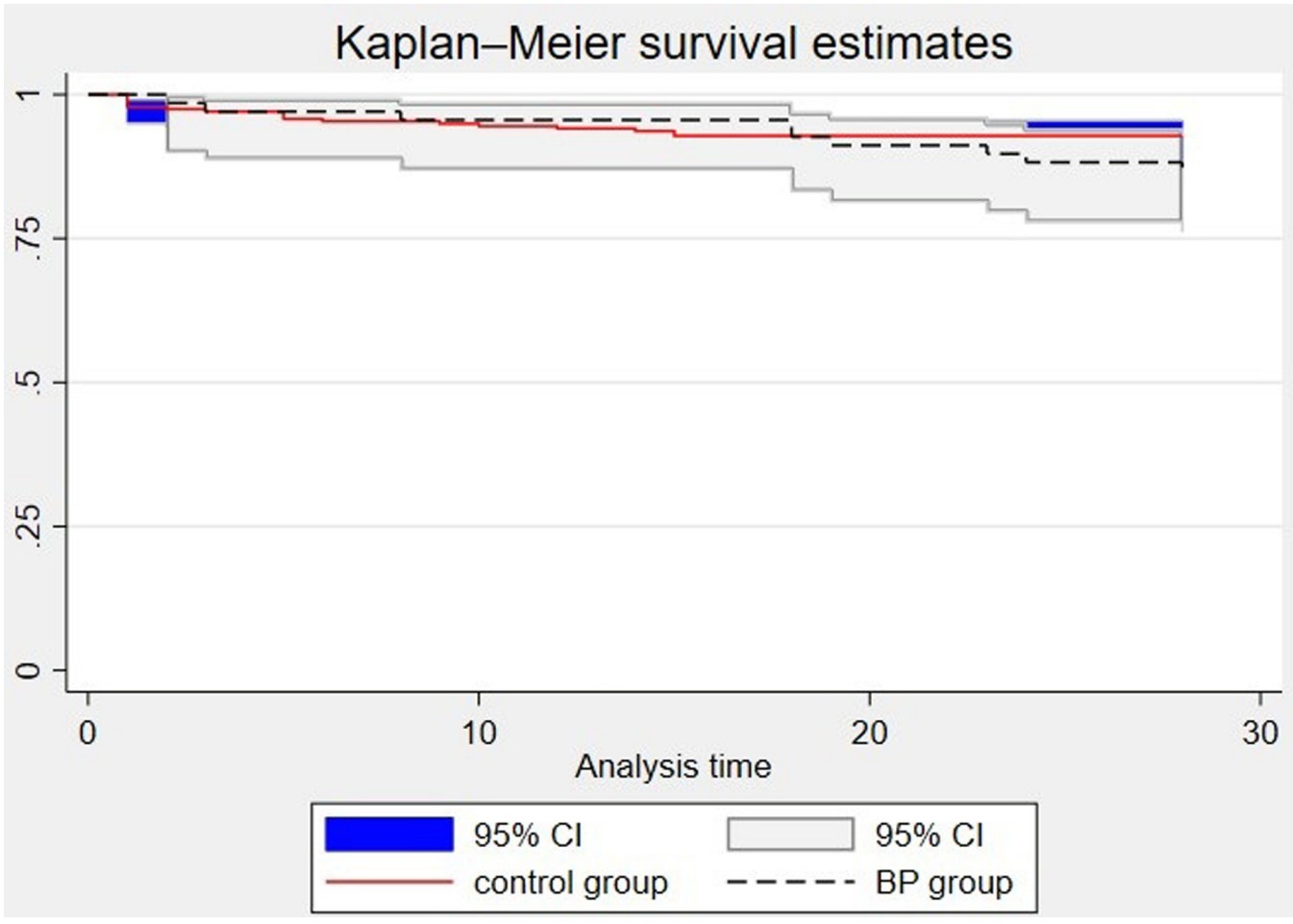

**Fig 3. 28-day Kaplan-Meier curve after PSM.**

literature search results. This finding suggests that Chinese clinicians may be more receptive to this technology given its potential benefits to patients.

Consistent with our previous results, age, the lowest lymphocyte count, and the lowest neutrophil count were still determined as risk factors for mortality of SAP patients in our study. With the progress of modern medicine, more ICU-admitted patients are able to survive the acute phase and progress to a chronic inflammatory state [21]. A study by Boomer et al. [22] confirmed the existence of a chronic inflammatory phase via analysis of tissue histology, inflammation characteristics, cell types, and cell surface molecules of spleen and lung tissue collected from patients who died from sepsis. The chronic inflammatory phase is mainly characterized by immunosuppression with impaired neutrophil function, reduced lymphocyte counts, decreased production of inflammatory mediators, increased PD-1, and decreased expression of CD69 and CD127 [23]. Warny et al. [24] found in a survey of nearly 100,000 samples that lymphopenia is an independent predictor of infection and death, with a 1.7-fold increase in infection-related mortality in patients with lymphopenia. An imbalanced immune response is a major contributor of SAP and even death [25]. Our previous data indicated that uncontrolled infections accounted for 34 out of 51 deaths within 2 weeks of SAP onset and that the death group had a lower lymphocyte count than the survival group [0.46 IQR (0.29–0.75) *vs.* 0.73 IQR (0.58–1.03), $p < 0.001$].

Whether BP treatment impacts lymphocytosis, neutrophil and monocyte counts and their function in pancreatitis patients is one of the most crucial unanswered questions. Research in this field is currently scarce, with the majority of studies concentrating on the high-volume hemofiltration (HVHF) model between 2000 and 2010. Peng et al. [26] went on to propose the immune regulation hypothesis of BP therapy for sepsis as a means to bridge this information gap. Furthermore, Yekebas et al. [27] found that HVHF could upregulate the expression of monocyte human leukocyte antigen DR (HLA-DR) and enhance neutrophil respiratory burst function in pigs with AP, which was accompanied by an increased survival rate in experimental animals for 60 days. The study also showed that the beneficial effect of HVHF is dose-dependent and associated with filter replacement.

It is reported that early HVHF (80 ml/kg/h) therapy (within 1–3 days after onset of illness) can downregulate the expression of Th-1 cytokines [interleukin (IL)-1, tumor necrosis factor-α (TNF-a), etc.], upregulate the expression of Th-2 cytokines (IL-10), and increase the expression of monocyte HLA-DR, in AP patients without sepsis [28]. This beneficial effect can be observed as early as 24 h after HVHF treatment. As demonstrated by elevated expression of monocyte HLA-DR, HVHF therapy administered to sepsis patients between 4 and 45 days into the illness can restore immune function in the late stages of pancreatitis. It can also enhance the expression of Th-1 cytokines in cultured lymphocytes and reduce the expression of Th-2 cytokines. However, reversed immune tolerance is only observed after 72 h of HVHF and is not as significant as in patients without sepsis. A clinical trial [29] including 12 patients with pancreatitis also showed that early HVHF performed over 3 days increased the number of peripheral blood monocytes, $CD4^+$, $CD8^+$ lymphocytes, and enhance monocyte HLA-DR expression. It also downregulated the concentration of Th-1 cytokines and increased the concentration of Th-2 cytokines when compared to standard treatment. In another study by Rokyta et al. [30], they observed the opposite, they found that CVVH did not improve HLA-DR expression in critically ill patients. These contradicting results from these two studies may be explained by the different BP methods and treatment doses, with the latter using the CVVH mode and a replacement fluid of 20–30 ml/kg/h.

Our research revealed that the BP group experienced a lower incidence of shock than the control group, which is in line with a prior study [31] that demonstrated how BP can stabilize septic patients' hemodynamics. The development of septic shock in SAP is caused by the necrotizing pancreatic tissue [32]. The tissue may become infected by bacteria with the usual culprits being *Staphylococcus aureus*, but mainly *Pseudomonas* and *Escherichia coli*. Sepsis is usually treated with Carbapenems (imipenem/cilastatin, meropenem, doripenem, biapenem or panipenem) [4] and vasopressor therapy [33]. However, sepsis-associated mortality was not improved with the use of vasoactive drugs. This could be due to the insignificant role of low blood pressure in the pathogenesis of circulatory shock. Indeed, emerging evidence has indicated that the damage caused by septic shock is mainly due to defective mitochondrial oxygen utilization rather than low blood pressure, leading to the hypothesis that shock is more likely to be the result rather than the cause of cell damage [34].

Low platelet count is an independent predictor of death [35, 36]. Of note, BP therapy filters, circuits, and heparin anticoagulation can lead to decreased platelet counts [37]. Our data indicate that the lowest platelet count in the death group was lower than that in the survival group [65.5 (IQR 33–115) *vs*. 152 (IQR 99–181), $p < 0.001$], and the lowest platelet count in the BP group was also lower than that in the control group [167 (IQR 138–197) *vs*. 96 (IQR 60–148), $p < 0.001$], suggesting that BP may have an adverse effect on platelet counts.

Overall, the effect of BP on mortality-associated risk factors such as age, gender, gastrointestinal ulcers, lymphocyte, and neutrophil counts and function has been established. Unfortunately, BP increases platelet destruction, indicating that BP has no beneficial effect on the

mortality of SAP patients. Our data also confirm that, after matching baseline characteristics, the mortality rate in the BP group did not improve.

In summary, the pathogenesis of AP remains undetermined, and the evidence supporting the use of BP as a treatment for inflammatory illnesses is still in the theoretical or animal experimentation stages. The results of our literature meta-analysis, as well as domestic and foreign single-center data analysis, collectively demonstrate that BP, except for HVHF mode, does not improve the survival rates of patients with SAP, nor does it reduce hospitalization time, lower costs, or mitigate the risk of local complications. The complexity of BP technology is due to the combination of various parameters. While it is possible that certain BP modes may provide survival benefits to specific subtypes of SAP patients, accurately identifying the appropriate target population poses a significant challenge. When using this technology to treat SAP, care should be taken until effective BP therapies can be developed and implemented, especially for patients who do not require RRT

This study has several limitations that should be noted. Firstly, despite using the PSM method, it is still a single-center retrospective case-control study, and confounding factors cannot be entirely eliminated. Secondly, the number of cases included is relatively small, and conducting subgroup analysis further decreases the sample size, which may result in unstable data for certain results. Large-scale, randomized, blinded, multicenter clinical trials are still required to evaluate BP's efficacy in treating SAP and to identify the possible patient populations that could benefit from this course of action from this treatment.

## Author Contributions

**Conceptualization:** Menghua Chen, Ken Okamoto.

**Formal analysis:** Hongwei Huang, Zhongshi Huang.

**Methodology:** Hongwei Huang, Zhongshi Huang.

**Software:** Hongwei Huang, Zhongshi Huang.

**Writing – original draft:** Hongwei Huang.

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
