## [Decision Letter · Decision Letter 0]

13 Nov 2023

PONE-D-23-23277Evaluation of the Therapeutic Efficiency and Efficacy of Blood Purification in the Treatment of Severe Acute PancreatitisPLOS ONE

Dear Dr. Okamoto,

Thank you for submitting your manuscript to PLOS ONE. After careful consideration, we feel that it has merit but does not fully meet PLOS ONE’s publication criteria as it currently stands. Therefore, we invite you to submit a revised version of the manuscript that addresses the points raised during the review process.

We look forward to receiving your revised manuscript.

Kind regards,

Chiara Lazzeri

Academic Editor

PLOS ONE

Journal Requirements:

**Additional Editor Comments:**

the present investigation deals with a hot topic. It is well designed however some issues can be rasied.

We suggest to add information on patients with septic shock, on the causes of septic shock (in particular microorganisms involved), antibiotics used (and wether serum concentrations varied between the two subgroups).

Mechanism(s) responsible for the different incidence of AKI should be hypothesized.

An English mother-tongue expert should revise the manuscript

Reviewers' comments:

Reviewer's Responses to Questions

**Comments to the Author**

1. Is the manuscript technically sound, and do the data support the conclusions?

Reviewer #1: Partly

Reviewer #2: Yes

2. Has the statistical analysis been performed appropriately and rigorously? 

Reviewer #1: I Don't Know

Reviewer #2: Yes

3. Have the authors made all data underlying the findings in their manuscript fully available?

Reviewer #1: Yes

Reviewer #2: Yes

4. Is the manuscript presented in an intelligible fashion and written in standard English?

Reviewer #1: No

Reviewer #2: Yes

5. Review Comments to the Author

Reviewer #1: 1.Is the unit of creatinine level “mmol/L” or“umol/L”

2.Please add the cause of pancreatitis like “Gallstones,Alcohol abuse,Hyperlipidemia” in the baseline character.

3.In table 3 there was statistically significant difference in 1-year all-cause mortality between two groups(P=0.04).What is the 1-year survival curves between two groups?

4 The English in this paper should be improved. I recommend the authors have it checked by a native speaker of English familiar with this field.

Reviewer #2: Overall, the data presented in the manuscript supports the similar findings described by other groups demonstrating limited effects of hemofiltration (ultrahemofiltration/dialysis) on the clinical course and outcomes of SIRS, sepsis, severe acute pancreatitis, etc. The main drawback of the paper that it is not really an original research but rather a statistical analysis of the retrospective data from Medical Information Mart for Intensive Care IV (MIMIC IV) database

provided by Beth Israel Deaconess Medical Center (Boston, Massachusetts, USA). Based on the methodology and the content of the paper, I believe, this manuscript should be targeted towards more specialized journal with the lower impact factor. I would also suggest changing the term "blood purification" into an appropriate medical term - hemodialysis, veno venous hemofiltration, etc.

6. PLOS authors have the option to publish the peer review history of their article (what does this mean?). If published, this will include your full peer review and any attached files.

Reviewer #1: No

Reviewer #2: No

---

## [Author Response · Author response to Decision Letter 0]

5 Dec 2023

Response to Reviewers

December 4, 2023

RE: Manuscript Number PONE-D-23-23277

Dear editor,

Thank you for granting us the opportunity to submit a revised version of the manuscript “Evaluation of the Therapeutic Efficiency and Efficacy of Blood Purification in the Treatment of Severe Acute Pancreatitis”. Thanks to the reviewers and editors for their thoughtful comments on our manuscript.

The suggested modifications are presented in the manuscript in trace mode, and a clean version was submitted without any revision marks. The reviewers' and editorial remarks and recommendations are also addressed below point by point. In addition, we also asked professional native editors to polish the language of our article.

Once again, we are grateful to the reviewers and editors for their valuable comments and suggestions and look forward to your response.

Please do not hesitate to contact us with any questions.

Sincerely,

Ken Okamoto

Point-by-point reply to Reviewers' comments

Reviewers' comments:

Reviewer #1:

1. Is the unit of creatine level “mmol/L” or “μmol/L”?

Response: Thank you for pointing this out. We agree with the Reviewer and have made changes, we have used μmol/L. See lines 86, 97, and 99.

2. Please add the cause of pancreatitis like “Gallstones, Alcohol abuse, Hyperlipidemia” in the baseline character.

Response: Thank you for this suggestion. Demographic information on etiology of AP is added to the Methods section, see line 106. The relevant expression “The Etiology of AP included cholelithiasis (72 cases), alcohol abuse (158 cases), hypertriglyceridemia (64 cases), and others (11 cases).” has been added to the baseline characteristics of the Results part, see lines 147-148, and see Table 1 for specific data.

3. In table 3 there was statistically significant difference in 1-year all-cause mortality between two groups (P=0.04).What is the 1-year survival curves between two groups?

Response: Thank you for pointing this out. There is no significant statistical difference in both the 28-day and 1-year all-cause mortality between the two groups. The value from 1 year was a simple oversight, this issue has been rectified (p = 0.114) instead of (p = 0.04). See Table 3.

4. The English in this paper should be improved. I recommend the authors have it checked by a native speaker of English familiar with this field.

Response: Thank you for this suggestion. This article has been carefully revised by professional native English editors to improve the language level and readability.

Reviewer #2:

Overall, the data presented in the manuscript supports the similar findings described by other groups demonstrating limited effects of hemofiltration (ultrahemofiltration/dialysis) on the clinical course and outcomes of SIRS, sepsis, severe acute pancreatitis, etc. The main drawback of the paper that it is not really an original research but rather a statistical analysis of the retrospective data from Medical Information Mart for Intensive Care IV (MIMIC IV) database provided by Beth Israel Deaconess Medical Center (Boston, Massachusetts, USA). Based on the methodology and the content of the paper, I believe, this manuscript should be targeted towards more specialized journal with the lower impact factor. I would also suggest changing the term "blood purification" into an appropriate medical term - hemodialysis, veno venous hemofiltration, etc.

Response: Thank you for your suggestion. Hemodialysis and hemofiltration are blood purification techniques (PMID: 37388799). In this paper, we explored the overall effect of blood purification on pancreatitis, as the patients used for this study underwent different types of blood purification treatments such as intermittent hemodialysis, continuous veno-venous hemofiltration, and an unknown form of BP (lines 216-217). Therefore blood purification was appropriately used to encompass all the different methods. Furthermore, many published articles use the term blood purification in their studies similar to our own (PMID: 30896634). (PMID: 35791651) lists hemofiltration as a type of blood purification treatment.

Point-by-point reply to editorial comments:

The present investigation deals with a hot topic. It is well designed however some issues can be rasied.

1. We suggest to add information on patients with septic shock, on the causes of septic shock (in particular microorganisms involved), antibiotics used (and wether serum concentrations varied between the two subgroups).

Response: Thank you for your suggestion. The relevant information (lines 294-298) has been added to the manuscript, as follows:

The development of septic shock in SAP is caused by the necrotizing pancreatic tissue [32]. The tissue may become infected by bacteria with the usual culprits being Staphylococcus aureus, but mainly Pseudomonas and Escherichia coli. Sepsis is usually treated with Carbapenems (imipenem/cilastatin, meropenem, doripenem, biapenem or panipenem) [4] and vasopressor therapy [33].

The references involved are as follows:

4. Mao E. Intensive management of severe acute pancreatitis. Ann Transl Med. 2019; 7(22): 687. doi: 10.21037/atm.2019.10.58

32. Mifkovic A, Pindak D, Daniel I, Pechan J. Septic complications of acute pancreatitis. Bratisl Lek Listy 2006; 107(8): 296-313.

33. Hamzaoui O, Goury A, Teboul JL. The Eight Unanswered and Answered Questions about the Use of Vasopressors in Septic Shock. J Clin Med 2023; 12(14): 4589. doi: 10.3390/jcm12144589

In addition, the data from this study was obtained from a publicly available database. The data did not explicitly state that the patients developed septic shock, we surmised that they had shock based on the use of vasopressors.

2. Mechanism(s) responsible for the different incidence of AKI should be hypothesized.

Response: Thank you for this suggestion. The patients from the control group had a lower comorbidity index and lower incidence of AKI than the group receiving BP treatment, the patients in the BP group usually had a more advanced SAP hence the need for BP treatment, and AKI is a major complication of SAP. The relevant contents (lines 237-242) are as follows:

About 64.7% of the BP group had chronic kidney disease (compared to 22.4% in the control group), and 30 patients had an acute worsening of chronic kidney function (defined as a serum creatinine increase of more than 1.5 times above the baseline) underwent BP treatment (24 people did not receive BP). Presumably, AKI is the main complication of SAP [20], and renal dysfunction or deterioration may lead to the requirement of BP treatment application in these patients.

The reference involved is as follows:

20. Kinjoh K, Nagamura R, Sakuda Y, Yamauchi S, Takushi H, Iraha T, Idomari K. Clinical efficacy of blood purification using a polymethylmethacrylate hemofilter for the treatment of severe acute pancreatitis. Acute Crit Care 2022; 37(3): 398-406. doi: 10.4266/acc.2022.00192

It should be noted that there is a large difference in the baseline between the two groups. See lines 163-164 and 235-237.

3. An English mother-tongue expert should revise the manuscript.

Response: Thank you for this suggestion. We have requested help regarding this issue and hopefully, it will be rectified.

---

## [Editor Report · Decision Letter 1]

18 Dec 2023

Evaluation of the Therapeutic Efficiency and Efficacy of Blood Purification in the Treatment of Severe Acute Pancreatitis

PONE-D-23-23277R1

Dear Dr. Okamoto,

We’re pleased to inform you that your manuscript has been judged scientifically suitable for publication and will be formally accepted for publication once it meets all outstanding technical requirements.

Kind regards,

Chiara Lazzeri

Academic Editor

PLOS ONE
---

## [Editor Report · Acceptance letter]

28 Dec 2023

PONE-D-23-23277R1 

PLOS ONE

Dear Dr. Okamoto, 

I'm pleased to inform you that your manuscript has been deemed suitable for publication in PLOS ONE. Congratulations! Your manuscript is now being handed over to our production team.

Kind regards, 

on behalf of

Dr. Chiara Lazzeri 

Academic Editor

PLOS ONE